Using stiffness to assess injury risk: comparison of methods for quantifying stiffness and their reliability in triathletes

Lorimer Anna V. alorimer@bond.edu.au avlorimer@gmail.com 1 2
Keogh Justin W.L. 1 2 3
Hume Patria A. 2
1 Faculty of Health Sciences and Medicine, Bond University , Gold Coast , Queensland , Australia
2 Sports Performance Research Institute New Zealand, Auckland University of Technology , Auckland , New Zealand
3 Faculty of Science, Health, Education and Engineering, University of the Sunshine Coast , Sunshine Coast , Queensland , Australia
Delahunt Eamonn
Electronic publication date: 2018 Oct 30
Publication date: 2018
Volume: 6
Electronic Location ID: e5845
Received 2018 Mar 12; Accepted 2018 Sep 28
Copyright: ©2018 Lorimer et al.
Copyright year: 2018
Copyright holder: Lorimer et al.
License: This is an open access article distributed under the terms of the Creative Commons Attribution License, which permits unrestricted use, distribution, reproduction and adaptation in any medium and for any purpose provided that it is properly attributed. For attribution, the original author(s), title, publication source (PeerJ) and either DOI or URL of the article must be cited.
License URL: https://creativecommons.org/licenses/by/4.0/

Keywords: Biomechanics, Running, Hopping, Joint, Leg, Vertical

Funding: Auckland University of Technology Vice Chancellors PhD scholarship Auckland University of Technology funded this research. Anna Lorimer was funded by a Vice Chancellors PhD scholarship. The funders had no role in study design, data collection and analysis, decision to publish, or preparation of the manuscript.

==============================
Background

A review of the literature has indicated that lower body stiffness, defined as the extent to which the lower extremity joints resists deformation upon contact with the ground, may be a useful measure for assessing Achilles injury risk in triathletes. The nature of overuse injuries suggests that a variety of different movement patterns could conceivably contribute to the final injury outcome, any number and combination of which might be observed in a single individual. Measurements which incorporate both kinetics and kinematics (such as stiffness) of a movement may be better able to shed light on individuals at risk of injury, with further analysis then providing the exact mechanism of injury for the individual. Stiffness can be measured as vertical, leg or joint stiffness to model how the individual interacts with the environment upon landing. However, several issues with stiffness assessments limit the effectiveness of these measures to monitor athletes’ performance and/or injury risk. This may reflect the variety of common biomechanical stiffness calculations (dynamic, time, true leg and joint) that have been used to examine these three stiffness levels (vertical, leg and joint) across a variety of human movements (i.e. running or hopping) as well as potential issues with the reliability of these measures, especially joint stiffness. Therefore, the aims of this study were to provide a comparison of the various methods for measuring stiffness during two forms of human bouncing locomotion (running and hopping) along with the measurement reliability to determine the best methods to assess links with injury risk in triathletes.

Methods

Vertical, leg and joint stiffness were estimated in 12 healthy male competitive triathletes on two occasions, 7 days apart, using both running at 5.0 ms−1 and hopping (2.2 Hz) tasks.

Results

Inter-day reliability was good for vertical (ICC = 0.85) and leg (ICC = 0.98) stiffness using the time method. Joint stiffness reliability was poor when assessed individually. Reliability was improved when taken as the sum of the hip, knee and ankle (ICC = 0.86). The knee and ankle combination provided the best correlation with leg stiffness during running (Pearson’s Correlation = 0.82).

Discussion

The dynamic and time methods of calculating leg stiffness had better reliability than the “true” method. The time and dynamic methods had the best correlation with the different combinations of joint stiffness, which suggests that they should be considered for biomechanical screening of triathletes. The knee and ankle combination had the best correlation with leg stiffness and is therefore proposed to provide the most information regarding lower limb mechanics during gait in triathletes.

Introduction

Overuse injuries are characterised by progressive onset of symptoms with no specific causal event. The causative mechanisms of overuse injuries have been suggested to be related externally to training loads and equipment or internally due to the biomechanics of the individual (Kannus, 1997). The progressive nature of overuse injuries suggests an accumulation over time of tissue damage that alone goes unnoticed and heals given sufficient time (Lorimer & Hume, 2014). Once damage reaches a tipping point however, pain and dysfunction result (Lorimer & Hume, 2014). Achilles tendon injuries in triathletes conceivably follow this pattern of tendon tissue micro damage, accumulated over time resulting in a progressively weakened tendon which is more likely to maintain further damage (Lorimer & Hume, 2014). Research into overuse injuries tends to isolate individual components of a movement to track risk factors. An accumulation of sub-threshold tissue challenges has been proposed as the mechanism of injury, suggesting that individual movement components that may predispose an athlete to injury may be too small to detect. It has been shown in a number of human movements, that similar final outcomes can be achieved through a variety of coordination strategies (Hiley, Zuevsky & Yeadon, 2013; Whiteside et al., 2015). Likewise, it is posited that progressive injuries have a variety of different contributors, any combination of which can lead to the final pain and dysfunction outcome.

It is believed that the study of movement patterns combining both kinetics and kinematics into single risk factor measures may provide greater insight into risk of injury than isolated movement components. Stiffness, of the lower limb ‘springs’ provides a means of assessing the influence of kinetics and kinematics on the tissues of the lower limb during running. Review of the literature indicated that five risk factors were associated with risk of Achilles injuries in running athletes (Lorimer & Hume, 2014; Lorimer & Hume, 2016). All five of these risk factors; surface stiffness, arch height, peak braking force, peak propulsive force and peak vertical force, linked to changes in lower limb stiffness (Lorimer & Hume, 2014; Lorimer & Hume, 2016).

Running as a bouncing gait has been modelled using the spring-mass model, a point mass balanced on a massless, compressible, linear spring (Blickhan, 1989; Günther & Blickhan, 2002; McMahon & Cheng, 1990; Seyfarth et al., 2002; Seyfarth, Geyer & Herr, 2003). The stiffness measure provides an holistic view of how the body accommodates the impacts of running, with the contributions of joints, muscles, tendons, ligaments and bones and the range of motion all contributing to the stiffness output (Butler, Crowell III & Davis, 2003). Stiffness therefore provides a useful tool to investigate how the combination of various movement components may combine to create an injurious environment (Lorimer & Hume, 2016).

The basic model of lower extremity stiffness has been represented in a number of mathematical forms (Fig. 1). Vertical stiffness is the most generalised, modelling the body as a point mass perched on top of a compressible, massless spring with compression of the spring represented by a negative vertical displacement of the centre of mass (McMahon & Cheng, 1990). Vertical stiffness can be measured using force platforms and can be estimated from flight time and contact time making this measurement appropriate for assessment of gait in a natural environment (Morin et al., 2005).

Figure 1 Biomechanical stiffness models.

(A) McMahon & Cheng’s (1990) spring-mass model for vertical and leg stiffness (McMahon & Cheng, 1990); (B) Coleman et al. (2012) true leg stiffness model; (C) joint stiffness model (Blickhan, 1989; Farley & Morgenroth, 1999).

The vertical displacement of the centre of mass used in calculations for vertical stiffness occurs due to changes in hip, knee and ankle angles, resulting in the leg becoming effectively shorter. Modelling leg stiffness where the leg is a spring, and there is compression of the ‘leg spring’ in response to contact with the ground, estimates stiffness taking into account the angle of leg swing (from ground contact to mid-stance), accounting for horizontal movement (Blickhan, 1989; McMahon & Cheng, 1990). During hopping, horizontal velocity is virtually zero, therefore leg stiffness is in essence equal to vertical stiffness (Farley et al., 1991). As anterior posterior braking force was determined to have a clear impact on Achilles injury risk, some incorporation of horizontal forces into the model should be considered (Coleman et al., 2012; Lorimer & Hume, 2014).

The compression of the ‘leg spring’ is achieved by rotation around the hip, knee and ankle joints, to dissipate landing forces and store energy for elastic return in the next step or hop. The individual joints have been modelled as rotational springs which have associated stiffness due to muscular activity controlling the speed and magnitude of the joint rotations, and mechanical behaviour of tendons, ligaments and other structures comprising the joints (Butler, Crowell III & Davis, 2003). Joint stiffness therefore provides insight into the body’s control strategies to attenuate the impact forces on landing and to the relative loading at each individual joint. When it comes to understanding overuse injuries, it is unclear which joint injurious mechanics can be attributed to or whether it is the interaction between the joints or extent to which joints compensate for each other which is important in understanding risk. In the support moment concept proposed by Winter (1980) it is noted that both the hip and knee have highly variable moments within and between individuals even when a similar support moment and gait velocity are maintained. It is possible, that looking at various combinations of the joints may provide insight into athletes relative risk of overuse injuries around particular lower body joints such as the Achilles tendon.

Triathletes and other endurance running athletes typically sustain near constant running velocities over a large portion of their training and races. The ability to maintain running velocity may require relatively consistent (reliable) vertical and leg stiffness. However, as vertical and leg stiffness may be controlled by the coordinated action at the ankle, knee and hip during the first half of stance to absorb impact and store energy for return during toe off, subtle differences in the body positions at foot strike may require step to step adjustments at the level of the ankle, knee and hip joints, thereby altering the individual joints stiffness. Similar to other biomechanical outputs that demonstrate that the variability of measurements may become greater as the measurement becomes more specific or isolated (Bartlett, Wheat & Robins, 2007; Stergiou & Decker, 2011), it is therefore likely that joint stiffness may be less stable (reliable) than leg or vertical stiffness. Such observations are supported by the reliability work investigating stiffness in single leg hopping (Diggin, Anderson & Harrison, 2016; Joseph et al., 2013) and overground running (Joseph et al., 2013). When comparing different hopping frequencies, the joints with the largest range of motion and therefore greatest potential for variability, the hip and knee had weak reliability (Diggin, Anderson & Harrison, 2016). When hopping is performed at the reported natural frequency of 2.2 Hz, reliability of the knee stiffness is improved, however hip reliability still remains poor (Diggin, Anderson & Harrison, 2016). The reliability of leg stiffness during hopping is generally reported to be good in populations of mixed athletes (Diggin, Anderson & Harrison, 2016; Joseph et al., 2013; Pruyn, Watsford & Murphy, 2016).

Another issue affecting the potential applicability of using a variety of stiffness measures to monitor performance and injury risk concerns the challenges in assessing running in ecologically valid situations. Currently, lower extremity stiffness (vertical, leg and joint) have been measured for over ground running and treadmill running as well as hopping, with the method of measurement largely dictated by space and equipment (Farley et al., 1991; Farley & Gonzalez, 1996; Morin et al., 2005). In order to measure ground reaction force for over ground running, targeting of a force platform is required, which has been suggested to alter aspects of the gait cycle and the forces associated with the individual foot strike (Challis, 2001). Various methods have been adopted to minimise the targeting effect, but the number of steps is also limited and it is difficult to control the running speed with overground running (Kluitenberg et al., 2012). Treadmill running allows the runner to adopt a more steady state biomechanics with the control of speed (Riley et al., 2008). The development of force plates built into treadmills allows the measurement of the ground reaction force for multiple consecutive steps in a confined space. Comparison of kinetics and kinematics for overground and treadmill running suggest treadmill running is comparable but not directly equivalent to overground running (Riley et al., 2008). Due to space constraints and the inability to record a sufficient number of foot strikes, hopping has also been used as a surrogate for running, as they both are bouncing gaits and requiring less space and technology to measure than overground or treadmill running. Hopping is usually conducted at a rate of 2.2 Hz (Farley et al., 1991) suggested as the natural hopping frequency. While a relationship between hopping stiffness and running performance has been well established (Butler, Crowell III & Davis, 2003; Chelly & Denis, 2001), a direct relationship between hopping and running stiffness has not, to the authors’ knowledge, been investigated in a triathlete population. As triathletes are not typically a group of athletes with much experience in hopping, it is unclear if the relationship reported in the literature between hopping and running stiffness for athletes who regularly run, and jump would also be observed in triathletes. This is an important consideration if hopping was to be used as a surrogate for running for injury risk factor analysis in triathletes.

To investigate whether stiffness can be used routinely to monitor running performance and injury risk, the reliability of the various measures of stiffness needed to be assessed. Where increased variability is present due to the inherent nature of human movement, understanding the variability that can be expected and to what extent the variability is minimised elsewhere in the kinetic chain is fundamental to understanding the results. It is also important to quantify the comparability of the various methods of measuring stiffness to determine the most appropriate measures to use for an investigation and to provide insight into the mechanism for any stiffness changes that may be observed.

The complexity of the human neuromusculoskeletal system allows for a relatively infinite combination of joint moment and activation patterns to be utilised when performing repeated activities, such as running and jumping. Joint compensation is supported by the observation of stable ground reaction forces and joint kinematics within subjects but highly variable hip and knee joint moments during both walking and running (Winter, 1980; Winter, 1984). For example, a decrease in ankle stiffness may be adjusted for by an increase in knee stiffness resulting in an overall more stable knee+ankle stiffness. It is important therefore to understand the reliability of combinations of joint stiffness as well as the reliability of the individual joints stiffness in isolation.

The aim of the study was to provide a comparison of the various methods for measuring stiffness during two forms of human bouncing locomotion (running and hopping) along with the measurement reliability to determine the best methods to assess links with injury risk in endurance athletes such as triathletes. Vertical and leg stiffness were hypothesised to show better reliability than the individual joints, even though the hip, knee and ankle all appear to play a role in the control of leg stiffness during running and hopping. Combining the joints in different combinations is hypothesised to reduce the measurement variability to allow interday assessment. Hopping is proposed to not be a good surrogate for running in triathletes when assessing injury risk.

Methods

Twelve well-trained, male triathletes (34 ± 5 y, 75.6 ±  6.2 kg, 1.80 ± 0.04 m) volunteered for the study. All triathletes were currently competitive as top level age group athletes in either Olympic or long distance events. Personal best times in the previous season of under 2 h 20 min for Olympic distance or under 10 h for Iron distance and the ability to run for over 2 min at 4.0 min/km was required. Triathletes were excluded if they currently had a lower limb injury or had not been back to full training for at least 6 weeks following a previous lower limb injury. To avoid the possible effects of maturation (Oliver & Smith, 2010) and ageing (Silder, Heiderscheit & Thelen, 2008; Strocchi et al., 1991), triathletes under 16 and over 50 were excluded. Triathletes were not excluded based on foot strike type to assess the utility of stiffness measures in a natural triathlon population. All triathletes provided fully informed written consent prior to participation. Ethical approval was obtained for all testing procedures from the Auckland University of Technology Ethics Committee (AUTEC reference #11/94).

A test–retest between-day protocol on two separate days, seven days apart was performed. Triathletes were requested to keep training the same in the week prior to each testing session (monitored using TrainingPeaks™, endurance load monitoring and coaching software) to minimise changes in running biomechanics which might occur because of fatigue due to different training volumes and intensities. Triathletes were not asked to refrain from exercise at any point to minimise disruption.

A treadmill graded run was performed initially on an instrumented treadmill, followed by bent knee and straight knee hopping tasks. Thermal effects on stiffness measures as a result of repeated energy dissipation are unknown, therefore tasks were not randomized. As the hopping task required practice, this was scheduled after the run trial. Running pace was not randomized to reduce the injury risk that may be associated with accelerating to the faster paces from an initial slow pace. Due to the racing level of the triathletes and the distances and speeds experienced in training, this protocol was unlikely to fatigue the triathlete and therefore non-randomisation was deemed acceptable (Abt et al., 2011). Following warm-up and familiarization of 5 min at 6.0 min/km (2.8 m/s), triathletes ran continuously for 2 min at each of 5.5, 5.0, 4.5 and 4.0 min/km (3.0, 3.3, 3.7 and 4.2 m/s). Cool down consisted of 1 min each at 5.5 min/km and 6.0 min/km. Acceleration and deceleration between running velocity blocks were set to 0.1 m/s2. In training, triathletes use pace to monitor running speed rather than horizontal velocity. Therefore, running pace was used to provide a familiar measure and to ensure that future injury outcome data can be accommodated to the triathletes training environment. Data were collected for the final 20 s of each 2 min block to ensure gait had stabilised following pace change. Triathletes were unaware of when recording was taking place.

After a 5 min rest, triathletes were given as much time as needed to familiarise themselves with hopping in time with the metronome. Triathletes performed single leg hopping, first on the right leg and then on the left, on the stationary treadmill (in-ground design with the treadmill belt in line with the laboratory floor). Hopping was carried out first with no instructions other than to keep in time to the metronome set at 2.2 Hz. Hopping was repeated with triathletes instructed to keep the knee as straight as possible. Ten hops were recorded once rhythm had stabilized to match the metronome frequency, based on visual inspection.

During the running and hopping tasks, all triathletes wore spandex shorts or trisuits and their own regular training shoes. Height (mm), mass (kg) and bilateral trochanterian height (mm) were recorded according to International Society for the Advancement of Kinanthropometry (ISAK) protocols (Stewart et al., 2011). Retroflective markers (10 mm) were attached to the lower body according to a modified three dimensional (3D) model (see Fig. 2) based on the models reported by Besier et al. (2003), Tulchin, Orendurff & Karol (2010) and Ferber, McClay Davis & Williams Iii (2003). Clusters of four markers, on thermo-moulded plastic shells were attached to the posterior pelvis (over the sacrum), anterior thigh (distal and lateral to avoid the bulk of muscle) and anterior shank (along the tibia). Anatomical markers were attached bilaterally to iliocristale, anterior superior iliac spine, trochanterion, medial and lateral femoral condyle, medial and lateral malleoli, proximal and distal calcaneus centre, the most anterolateral aspect on the distal border of the calcaneus, first and fifth metatarsal heads and centre line of the forefoot between the second and third metatarsal heads. Following a static standing calibration, femoral condyle and malleoli markers were removed. For dynamic calibration of the hip joint, participants moved first the right then the left leg through a combination of flexion, abduction, adduction and extension (Besier et al., 2003; Piazza, Okita & Cavanagh, 2001). Knee joint centre dynamic calibration involved three squat movements (Besier et al., 2003).

Figure 2 Lower body marker locations without and with tracking clusters.

Photo credit: Prof. Patria Hume.

A 9-camera VICON motion analysis system (Oxford Metrics Ltd., Oxford, UK) combined with a Bertec instrumented treadmill (BERTEC Corp, Worthington, OH, USA) were used for kinematic (200 Hz) and vertical, horizontal and lateral ground reaction force (1,000 Hz) collection, respectively. Full analysis of the Bertec instrumented treadmill force performance, both static and dynamic, has been reported by Belli et al. (2001). Vertical force maximal non-linearity was ±0.3% with a relative error of 0.11%. Force differences between the treadmill and an artificial walking leg ranged from ±4.6 to ±20.9 N. Functional joint positions were determined using a custom built, MATLAB constrained optimization program (Optimization Toolbox, Mathworks Inc.; Natick, MA, USA) detailed by Besier et al. (2003). Joint angles, moments and foot centre of pressure locations were calculated via inverse kinematics using Visual3D software (Visual 3D, C-motion, Inc.; Germantown, MD, USA). Anatomical co-ordinate systems were defined according to specifications reported by Besier et al. (2003). For the single segment foot, the x-axis was the line joining the two calcaneal markers. The y-axis followed the longitudinal axis of the foot from the proximal calcaneal marker, to the forefoot midline marker. The z-axis was orthogonal to the x and y axes.

Variables were averaged over ten steps per leg for each individual for the running trials to allow for step variability (Dalleau et al., 1998) taken from the first full step recorded in the last 20 s of each running block based on the lack of no significant difference in stride to stride differences for biomechanical measures (Dalleau et al., 2004; Morin et al., 2005). Five consecutive hops within 5% of the 2.2 Hz hopping frequency were averaged (Granata, Padua & Wilson, 2002). Horizontal velocity was taken as treadmill velocity and was assumed to be constant. Stiffness values were normalized to body mass before statistical analysis.

Stiffness values were calculated using a custom written Labview program (Labview, National Instruments Corp.; Austin, TX). Stiffness was calculated for the first half of stance from initial heel contact to maximal vertical ground reaction force for all stiffness measures (Joseph et al., 2013). Stiffness calculations were carried out using all the equations stated in Table 1. The method reported by Coleman et al. (2012) was used to calculate kleg∕brake as absolute change of force and leg length between ground contact and Fmax. The greater trochanter marker (GTR) was used as reported by Coleman et al. (2012) to give kleg∕GTR. Improved repeatability is reported from using functional hip and knee joints for defining the leg segments (Besier et al., 2003), therefore the functional hip joint centre (HJC) as the hip marker (kleg∕HJC) was compared to kleg∕GTR. Joint stiffness combinations, ksumjoints, khip+knee and kknee+ankle were calculated using Eqs. (1)–(3).

Table 1 Biomechanical stiffness model calculations, variables and equipment.

Equation	Stiffness calculation	Terms list	Key measures	
	Vertical stiffness			
V1	kvert∕dynamic=FmaxΔy	Fmax = peak vertical force, Δy = centre of mass displacement from double integration Fmax	Vertical force
(McMahon & Cheng, 1990)	
V2	kvert∕time=FmaxΔy
Fmax=mgπ2tftc+1
Δy=−Fmaxmtc2π2+gtc28	m = subject mass, tc = contact time, tf = flight time, g = acceleration due to gravity	Contact time (video or contact mat)
(Morin et al., 2005)	
	Leg stiffness			
L1	kleg∕dynamic = FmaxΔL
ΔL=Δy+L01−cosθ
θ=sin−1vtc2L0	Fmax = peak vertical force, ΔL = change in leg length, Δy = centre of mass displacement from double integration of force, L0= trochanterian height, θ = angle of leg swing, v horizontal velocity, tc = contact time	Vertical force
Horizontal velocity Standing leg length (McMahon & Cheng, 1990; Morin et al., 2005)	
L2	kleg∕time=FmaxΔLFmax=mgπ2tftc+1Δy=−Fmaxmtc2π2+gtc28ΔL=L0−L02−vtc22+Δy	Fmax = peak vertical force, ΔL = change in leg length, Δy = centre of mass displacement, L0= trochanterian height, m = subject mass, tc= contact time, tf= flight time, g = acceleration due to gravity, v= horizontal velocity	Contact time
Horizontal velocity Standing leg length (Morin et al., 2005)	
L3	kleg∕brake=maxFlegΔLtrue
Fleg=cosθlegFR
FR=Fv2+FH2
θleg=90−θtrue−θRθR=cos−1FvFRθtrue=tan−1AB	max Fleg = maximal force directed in line of the leg, ΔLtrue = true change in leg length, θleg = angle of leg, FR = resultant force, FV = vertical force, FH = horizontal force, θtrue = angle between the calculated Ltrue and horizontal axis, θR = angle of the resultant force, A = vertical distance from hip marker to ground, B = horizontal distance from hip marker to centre of pressure	Horizontal force
Vertical force
High speed video
Hip marker
Centre of pressure (Coleman et al., 2012)	
	Joint stiffness			
J1	kjoint=ΔMΔθ	ΔM = change in joint moment, Δθ = change in joint angle	Three dimensional force
Lower body video for inverse dynamics calculation (Farley & Morgenroth, 1999; Günther & Blickhan, 2002)	

(1) ksumjoints=khip+kknee+kankle

(2) khip+knee=khip+kknee

(3) kknee+ankle=kknee+kankle.

Descriptive statistics including group means and standard deviations were calculated for all measures for both hopping and running. Data were assessed for between trial measurement reliability and measurement variability at the 90% confidence level following log transformation to allow results to be expressed as percentages (Hopkins, 2000). Robustness was maintained by using two criteria each to determine the level of reliability and variability (Bradshaw et al., 2010).

Average reliability was determined to be ‘good’ when the percent difference between means (MDiff%) was <5% and the effect size (ES) was trivial (0–0.2) or small (0.2–0.6) (Hopkins et al., 2009). If one of these criteria were not met, then measurement reliability was interpreted as ‘average’. ‘Poor’ reliability meant neither criteria was met (Bradshaw et al., 2010).

Measurement variability was assessed from typical error, reported as coefficient of variation percentage (CV%) (Bradshaw et al., 2010; Hopkins, 2000) and intra-class correlation coefficient (ICC) with upper and lower confidence limits (Bradshaw et al., 2010). Criteria for ‘small’ measurement variability were CV <10% (Bradshaw et al., 2010) and ICC > 0.70 (Bradshaw et al., 2010; Hopkins et al., 2009). If CV was >10% or ICC < 0.70 then variability in the measurement was deemed ‘moderate’. ‘Large’ measurement variability was reported if neither criteria for ‘small’ was met.

For overall reliability, all four variables (effect size, percent difference between means, coefficient of variation and interclass correlation) were assessed. ‘Good’ reliability required all four criteria to be met. ‘Moderate’ reliability resulted from one criteria outside the limits, while if two or more criteria were outside the limits, a ‘poor’ overall reliability was recorded (Joseph et al., 2013).

Biomechanical stiffness models were checked for comparability using Pearson’s correlation coefficient both within the stiffness type and between stiffness types (vertical with leg stiffness and leg with joint stiffness). Leg stiffness and joint stiffness have different units, therefore results were converted to unitless values prior to comparison using Eq. (4) (Liew, Netto & Morris, 2017; McMahon & Cheng, 1990) and Eq. (5) (Rummel et al., 2008).

(4) Dkleg=klegl0mg

(5) Dkjoint=kjointmgl0

A Pearson’s correlation coefficient > 0.90 was interpreted to show ‘very large’ correlation between the stiffness models or stiffness types. Correlations between 0.70 and 0.90 indicated ‘large’ comparability, moderate was considered to have been obtained with correlations between 0.50–0.69 while anything below 0.50 indicated a poor correlation between the two variables of interest (Hopkins et al., 2009). Hopping data were compared to running data for all calculated variables using the above criteria.

Results

Data for the left leg only are presented in Tables 2 and 3 as both the right and left leg showed similar results across all variables. Only the 5.0 min/km (3.3 m/s) running pace was presented to keep the results concise, with all speeds showing similar trends. Running at 5.0 min/km was considered to be a speed that would be encountered in training and/or racing by both elite and amateur triathletes, based on discussion with athletes, coaches and High Performance Sport New Zealand. Descriptive statistics are presented in Table 2 for all variables analysed for the left leg during running (5.0 min/km), hopping with a natural knee bend and hopping with the knee as straight as possible.

Table 2 Average stiffness for running and hopping tasks in triathletes.

Method	Mean (±SD)	Method	Mean (±SD)	Method	Mean (±SD)	
Vertical-Run (kN/m/kg)	Leg–Run (kN/m/kg)	Joint–Run (Nm/∘/kg)	
kvert∕dynamic	0.34 ± 0.06	kleg∕brakeHJC	0.40 ± 0.10	ksumjoints	0.47 ± 0.10	
kvert∕time	0.36 ± 0.05	kleg∕brakeGTR	0.40 ± 0.11	kankle+knee	0.26 ± 0.05	
		kleg∕dynamic	0.15 ± 0.03	khip+knee	0.33 ± 0.08	
		kleg∕time	0.15 ± 0.02	kankle	0.14 ± 0.03	
				kknee	0.11 ± 0.04	
				khip	0.21 ± 0.07	
	Vertical/Leg–Bent Hop (kN/m/kg)	Joint–Bent Hop (Nm/∘/kg)	
		Hkleg∕brakeHJC	0.23 ± 0.03	Hksumjoints	0.51 ± 0.27	
		Hkleg∕brakeGTR	0.23 ± 0.03	Hkankle+knee	0.41 ± 0.22	
		Hkleg∕dynamic	0.21 ± 0.04	Hkhip+knee	0.45 ± 0.22	
		Hkleg∕time	0.22 ± 0.02	Hkankle	0.10 ± 0.01	
				Hkknee	0.17 ± 0.06	
				Hkhip	0.31 ± 0.18	
	Vertical/Leg–Straight Hop (kN/m/kg)	Joint–Straight Hop (Nm/∘/kg)	
		Hkleg∕brakeHJC	0.24 ± 0.02	Hksumjoints	0.43 ± 0.14	
		Hkleg∕brakeGTR	0.23 ± 0.02	Hkankle+knee	0.22 ± 0.06	
		Hkleg∕dynamic	0.22 ± 0.03	Hkhip+knee	0.33 ± 0.12	
		Hkleg∕time	0.23 ± 0.02	Hkankle	0.10 ± 0.03	
				Hkknee	0.13 ± 0.04	
				Hkhip	0.20 ± 0.10	

Table 3 Summary of reliability results with 90% confidence interval for various biomechanical stiffness models for running and hopping tasks in triathletes.

Method	Reliability	Variability	Overall reliability	
	Mdiff%	ES	CV%	ICC		
Running - Vertical	
kvert∕dynamic	3.5 (0.4–6.7)	−0.18 (−0.21–−0.15)	4.2 (3.2–6.7)	0.95 (0.88–0.98)	Good	
kvert∕time	0.1 (−3.5–3.8)	0.01 (−0.02–0.03)	5.2 (3.8–8.1)	0.85 (0.62–0.94)	Good	
Running - Leg	
kleg∕dynamic	0.3 (−1.8–2.4)	−0.00 (−0.01–0.01)	2.8 (2.0–4.4)	0.98 (0.93–0.99)	Good	
kleg∕time	0.3 (−1.8–2.4)	−0.00 (−0.01–0.01)	2.8 (2.0–4.4)	0.98 (0.93–0.99)	Good	
kleg∕HJC	3.2 (−3.7–10.5)	−0.11 (−0.17–−0.06)	9.8 (4.2–15.6)	0.85 (0.63–0.94)	Gooda	
kleg∕GTR	−1.4 (−9.2–6.9)	0.13 (0.09–0.18)	11.8 (8.7–18.8)	0.77 (0.46–0.91)	Moderatea	
Running - Joints	
ksumjoints	−5.4 (−10.5–0.0)	0.36 (0.32–0.40)	7.9 (5.8–12.5)	0.86 (0.65–0.95)	Moderatea	
kknee+ankle	−3.1 (−8.4–2.5)	0.18 (0.16–0.20)	8.0 (5.9–12.7)	0.87 (0.66–0.95)	Gooda	
khip+knee	−5.8 (−11.4–0.1)	0.33 (0.29–0.37)	8.7 (6.5–13.9)	0.88 (0.70–0.96)	Moderatea	
kankle	−4.2 (−10.2–2.3)	0.31 (0.30–0.33)	9.3 (6.9–14.8)	0.75 (0.42–0.90)	Gooda	
kknee	−1.3 (−10.2–8.4)	0.02 (0.00–0.05)	13.8 (10.1–22.1)	0.90 (0.73–0.96)	Moderate	
khip	−7.7 (−14.8–0.0)	0.43 (0.41–0.46)	11.6 (8.5–18.5)	0.83 (0.59–0.94)	Poora	
Bent Knee Hopping - Leg	
Hkleg∕dynamic	−7.0 (−11.6–−2.2)	0.34 (0.32–0.36)	5.9 (4.2–10.4)	0.92 (0.74–0.97)	Moderate	
Hkleg∕time	−3.7 (−6.6–−0.7)	0.51 (0.50–0.52)	3.9 (2.8–6.4)	0.73 (0.33–0.91)	Gooda	
Hkleg∕HJC	−5.5 (−10.1–−0.7)	0.43 (0.41–044)	6.3 (4.5–10.5)	0.79 (0.45–0.93)	Moderatea	
Hkleg∕GTR	−3.5 (−7.7–0.9)	0.38 (0.36–0.39)	5.5 (4.0–9.3)	0.71 (0.30–0.90)	Gooda	
Bent Knee Hopping - Joints	
Hksumjoints	−10.4 (−32.2–18.5)	0.03 (−0.11–0.17)	13.0 (9.1–23.2)	.52 (−0.07–0.83)	Poora	
Hkhip+knee	−10.4 (−35.4–24.3)	0.22 (0.10–0.35)	49.1 (33.8–92.9)	0.32 (−0.28–0.72)	Poora	
Hkknee+ankle	−4.1 (−32.0–35.1)	0.03 (−0.10–0.17)	48.0 (32.5–95.5)	0.39 (−0.23–0.78)	Poora	
Hkankle	−8.2 (−17.5–−2.1)	0.56 (0.55–0.57)	13.0 (9.1–23.2)	0.52 (−0.07–0.83)	Poora	
Hkknee	2.9 (−18.5–29.9)	−0.13 (−0.17–−0.08)	32.9 (23.1–59.7)	0.51 (−0.03–0.82)	Poora	
Hkhip	−31.9 (−51.5–−3.6)	0.59 (0.49–0.68)	48.1 (32.6–95.7)	0.56 (−0.03–0.85)	Poora	
Straight Knee Hopping - Leg	
Hkleg∕dynamic	2.8 (−1.9–7.7)	−0.18 (−0.20–−0.16)	5.8 (4.2–9.8)	0.91 (0.73–0.97)	Gooda	
Hkleg∕time	1.1 (−1.6–3.9)	−0.14 (−0.15–−0.13)	5.8 (4.2–9.8)	0.85 (0.62–0.95)	Gooda	
Hkleg∕HJC	4.2 (−1.5–10.2)	−0.26 (−0.27–−0.24)	7.1 (5.1–12.0)	0.69 (0.24–0.89)	Moderatea	
Hkleg∕GTR	6.2 (0.8–11.8)	−0.50 (−0.51–−0.48)	6.5 (4.7–10.9)	0.69 (0.25–0.89)	Poora	
Straight Knee Hopping - Joints	
Hksumjoints	−2.7 (−13.7–22.3)	−0.08 (−0.16–0.00)	23.7 (16.8–41.9)	0.66 (0.20–0.88)	Poora	
Hkhip+knee	1.7 (−17.4–23.3)	−0.10 (−0.17–−0.02)	28.9 (20.4–51.9)	0.63 (0.15–0.87)	Poora	
Hkknee+ankle	8.6 (−12.7–35.2)	−0.28 (−0.32–−0.23)	30.6 (21.5–55.2)	0.25 (−0.35–0.69)	Poor	
Hkankle	−0.8 (−10.8–10.4 )	0.09 (0.07–0.10)	14.8 (10.8–24.7)	0.76 (0.42–0.91)	Moderatea	
Hkknee	9.6 (−17.5–45.7)	−0.33 (−0.37–−0.29)	44.5 (31.3–79.8)	0.14 (−0.38–0.60)	Poor	
Hkhip	−6.6 (−22.6–12.7)	0.11 (0.06–0.17)	25.8 (18.2–45.9)	0.81 (0.49–0.93)	Poora	
Notes.

a At least one reliability parameter was unclear (confidence interval spanned more than one criteria).

Summary of results

Hopping with a straight knee resulted in a reduction in knee stiffness compared to bent knee hopping (natural hopping with no instruction) to give an average value closer to running knee stiffness. Ankle stiffness was lower in both hopping conditions compared to running and did not significantly differ between the two hopping conditions. The combined stiffness of the hip, knee and ankle was also less in straight knee hopping than running. All joint variables were closer to running for straight knee than bent knee hopping. Leg stiffness was more than two-fold greater when using the kleg∕brake (Equation L3) estimation than for kleg∕dynamic (Equation L1) or kleg∕time (Equation L2). However, the magnitude of the kleg∕brake (Equation L3) estimate was closer to the combined joint stiffness. Ankle stiffness was greater than knee stiffness during running but this relationship was reversed for both hopping conditions. The variation (% of mean) of knee stiffness was larger than the other two joints and this variation was greatest with straight leg hopping (60%). Variation of the stiffness measurements increased when moving from the “global” vertical stiffness to the more focused joint stiffness.

Reliability analysis

The reliability and measurement variability are reported in Table 3. All running vertical and leg stiffness variables showed good overall reliability except for the kleg∕brake (Equation L3) estimates. Good reliability was achieved when the top of the ‘true’ leg was measured from the modelled hip joint centre (kleg∕brakeHJC). When measured from the single trochanterion marker, overall reliability was moderate. Individual joint stiffness ranged from poor to good with the hip having the poorest and ankle having the best overall reliability. Combining the joints as hip, knee and ankle, hip and knee or knee and ankle tended to improve the reliability to between moderate and good. For bent knee hopping kleg∕time (Equation L2) and kleg∕brakeGTR (Equation L3) methods showed good reliability, with all other leg stiffness measures showing only moderate reliability. All joint stiffness estimates, including joint combinations had poor overall reliability for bent knee hopping. Straight leg hopping improved the kleg∕dynamic (Equation L1) estimate reliability but gave only poor to moderate kleg∕brake (Equation L3) reliability. Overall reliability of the ankle was improved with straight leg hopping.

Comparing hoping and running stiffness

Stiffness during hopping with the straight and bent knee condition were compared with the related running stiffness measurement (Table 4). Comparability of hopping and running stiffness was poor for all joint measures (Equation J) when hopping with a bent knee except the ankle which was moderate. All leg stiffness measures for straight knee hopping had poor comparability with running. Moderate comparability was achieved between bent knee hopping and running for all leg measurements except kleg∕brakeGTR (Equation L3) which was poor. Conversely, comparisons between straight knee hopping and running were moderate for the individual knee and ankle (Equation J), hip+knee (Eq. (2)) and sumjoints (Eq. (1)) and large for knee+ankle (Eq. (3)). Only the hip was poorly correlated with running.

Table 4 Comparison between stiffness measures for running at 5.0 min/km and hopping at 2.2 Hz in triathletes.

Method	Comparison with	Pearson’s correlation (90% CL)	Interpretation	
Hopping - Leg		
- Bent Knee		
Hkleg∕dynamic	Rkleg∕dynamic	0.66 (0.20–0.88)	Moderatea	
Hkleg∕time	Rkleg∕time	0.53 (0.01–0.82)	Moderatea	
Hkleg∕brakeHJC	Rkleg∕brakeHJC	0.55 (0.07–0.82)	Moderatea	
Hkleg∕brakeGTR	Rkleg∕brakeGTR	0.34 (−0.20–0.72)	Poora	
- Straight Knee		
Hkleg∕dynamic	Rkleg∕dynamic	0.37 (−0.19–0.75)	Poora	
Hkleg∕time	Rkleg∕time	0.36 (−0.20–0.74)	Poora	
Hkleg∕brakeHJC	Rkleg∕brakeHJC	0.01 (−0.49–0.51)	Poora	
Hkleg∕brakeGTR	Rkleg∕brakeGTR	0.02 (−0.48–0.52)	Poora	
Hopping - Joint		
- Bent Knee		
Hksumjoints	Rksumjoints	−0.17 (−0.62–0.36)	Poora	
Hkhip+knee	Rkhip+knee	−0.31 (−0.70–0.22)	Poora	
Hkknee+ankle	Rkknee+ankle	0.49 (−0.01–0.80)	Poora	
Hkankle	Rkankle	0.65 (0.23–0.87)	Moderatea	
Hkknee	Rkknee	0.49 (−0.01–0.79)	Poora	
Hkhip	Rkhip	−0.38 (−0.74–0.15)	Poora	
Hkknee+ankle	Hkleg∕time	0.46 (−0.12–0.81)	Poora	
- Straight Knee		
Hksumjoints	Rksumjoints	0.63 (0.19–0.86)	Moderatea	
Hkhip+knee	Rkhip+knee	0.61 (0.16–0.85)	Moderatea	
Hkknee+ankle	Rkknee+ankle	0.73 (0.35–0.90)	Largea	
Hkankle	Rkankle	0.54 (0.05–0.82)	Moderatea	
Hkknee	Rkknee	0.62 (0.18–0.86)	Moderatea	
Hkhip	Rkhip	0.47 (−0.03–0.79)	Poora	
Hkknee+ankle	Hkleg∕time	0.57 (0.06–0.84)	Moderatea	
Notes.

a Indicates confidence interval spans more than one interpretation bracket.

Comparing leg stiffness with vertical and joint stiffness

As the middle level of stiffness, leg stiffness estimates were compared with vertical and joint stiffness estimates (Table 5). The highest correlation with each of the combinations of joints, sumjoints (Eq. (1)), hip+knee (Eq. (2)), knee+ankle (Eq. (3)) (r = 0.61, r = 0.66, r = 0.82) was with kleg∕time (Equation L2). Confidence limits for all joint measurement correlations were large, with this variation smallest between kknee+ankle (Eq. (2)) and kleg∕time (Equation L2). The time (Equation L2) and dynamic (Equation L1) methods had large correlations with knee stiffness (Equation J), however ankle and hip stiffness alone showed poor correlations with leg stiffness. All kleg estimates had large correlations with both kvert∕dynamic(Equation V1) and kvert∕time (Equation V2).

Table 5 Comparison of vertical and joint stiffness measures with leg stiffness for treadmill running at 5.0 min/km.

Method	Comparison with	Pearson’s correlation (90% CL)	Interpretation	
Vertical				
kvert∕dynamic	kleg∕dynamic	0.89 (0.70–0.97)	Largea	
	kleg∕time	0.79 (0.45–0.93)	Largea	
	kleg∕brakeHJC	0.88 (0.68–0.96)	Largea	
	kvert∕time	0.83 (0.57–0.94)	Largea	
kvert∕time	kleg∕dynamic	0.87 (0.64–0.96)	Largea	
	kleg∕time	0.91 (0.73–0.97)	Very largea	
	kleg∕brakeHJC	0.87 (0.65–0.95)	Largea	
Leg				
kleg∕brakeGTR	kleg∕dynamic	0.68 (0.25–0.89)	Moderatea	
	kleg∕time	0.64 (0.18–0.87)	Moderatea	
	kleg∕brakeHJC	0.97 (0.91–0.99)	Very large	
kleg∕brakeHJC	kleg∕dynamic	0.72 (0.32–0.90)	Largea	
	kleg∕time	0.69 (0.27–0.89)	Moderatea	
kleg∕time	kleg∕dynamic	0.94 (0.83–0.98)	Very largea	
Joint				
ksumjoints	kleg∕brakeHJC	0.55 (0.06–0.82)	Moderatea	
	kleg∕brakeGTR	0.39 (−0.13–0.75)	Poora	
	kleg∕dynamic	0.49 (−0.05–0.81)	Poora	
	kleg∕time	0.61 (0.13–0.86)	Moderatea	
kknee+ankle	kleg∕dynamic	0.75 (0.37–0.91)	Largea	
	kleg∕time	0.82 (0.52–0.94)	Largea	
	kleg∕brakeHJC	0.69 (0.29–0.88)	Moderatea	
khip+kne	kleg∕dynamic	0.53 (0.00–0.82)	Moderatea	
	kleg∕time	0.66 (0.21–0.88)	Moderatea	
	kleg∕brakeHJC	0.54 (0.06–0.82)	Moderatea	
kankle	kleg∕dynamic	0.20 (−0.36–0.65)	Poora	
	kleg∕time	0.28 (−0.29–0.70)	Poora	
	kleg∕brakeHJC	0.39 (−0.14–0.74)	Poora	
kknee	kleg∕dynamic	0.76 (0.40–0.92)	Largea	
	kleg∕time	0.79 (0.46–0.93)	Largea	
	kleg∕brakeHJC	0.55 (0.08–0.83)	Moderatea	
khip	kleg∕dynamic	0.17 (−0.39–0.64)	Poora	
	kleg∕time	0.31 (−0.25–0.72)	Poora	
	kleg∕brakeHJC	0.32 (−0.22–0.70)	Poora	
BHkknee+ankle	BHkleg∕time	0.46 (−0.12–0.81)	Poora	
SHkknee+ankle	SHkleg∕time	0.57 (0.06–0.84)	Moderatea	
Notes.

BH bent knee hopping

SH straight knee hopping

a Indicates confidence interval spans more than one interpretation bracket.

Discussion

Understanding the mechanism of overuse injuries is confounded by the insidious nature of the injury with lack of a single defining injurious event. Biomechanical research into injuries has tended to focus on individual joints or muscles to isolate risk factors for injuries. It is possible that it is the interaction between multiple segments and structures within the kinetic chain that results in progressive disruption of the structure and function resulting in overuse injuries (Lorimer & Hume, 2014). The various levels of stiffness allow measurement of this interaction and may provide a useful first step in the analysis of the injury process for overuse injuries such as Achilles tendon injuries.

Analysis of joint stiffness gives greater information regarding the mechanics of the lower limb, however the poor reliability of isolated joint stiffness estimates (Joseph et al., 2013) has currently limited the usefulness of such measures. Compression of the ‘leg spring’ is achieved through rotation of the joints and therefore joint stiffness should be related to leg stiffness. The contribution of each joint to the overall leg stiffness needs to be established to understand how triathletes and other athletes adjust stiffness in response to changes in task or environment constraints. The use of hopping as a surrogate for running would limit the space and equipment requirements, however, the correlation between hopping stiffness and running stiffness first needs to be established in a variety of populations.

Mean kvert∕dynamic (Equation V1) was similar to results reported for treadmill running (Dutto & Smith, 2002). When contact and flight time were used to estimate vertical stiffness (kvert∕time, equation V2), stiffness was similar to kvert∕dynamic (Equation V1) but substantially smaller than the average stiffness reported by Hunter & Smith (2007) using the same time based calculation method. The lower stiffness in the current research could be the result of using different populations: competitive triathletes versus general runners. Alternatively, use of a measured leg length, in the current data, rather than estimated leg length could account for the differences in stiffness reported. Running speed was also different for each individual in the runners assessed by Hunter & Smith (2007) which may further contribute to these between study population differences which influences the vertical stiffness estimates (Brughelli & Cronin, 2008; Hunter & Smith, 2007).

Mean kleg∕dynamic (Equation L1) and kleg∕time (Equation L2) for the triathletes were similar to results for treadmill running (Dutto & Smith, 2002; Hunter & Smith, 2007) but were lower than for overground running for the respective calculations (Arampatzis, Bruggemann & Metzler, 1999; Coleman et al., 2012). Treadmill running tends to be more upright with a shorter stride, higher cadence and flatter foot contact than over ground running (Nigg, De Boer & Fisher, 1995; Riley et al., 2008). Increased cadence is associated with increased kleg (Dutto & Smith, 2002; Farley & Gonzalez, 1996; Girard, Micallef & Millet, 2011; Girard et al., 2013). However, reducing the contact angle of the stance leg may contribute to lower kleg (Seyfarth et al., 2002; Seyfarth, Geyer & Herr, 2003). Alternately, the requirements of landing on a force platform in overground running also introduces the issue of targeting which could alter joint angles and landing forces at contact, contributing to some difference in stiffness estimates than what would have occurred naturally in over ground running gait (Challis, 2001). Joseph et al. (2013) reported lower kleg∕dynamic (Equation L1) than the current data for over ground running in middle distance runners. Due to the small sample sizes in the present study and those studies reported above, population differences would alter the mean stiffness estimates (Hopkins, 2006). While triathletes do not have significantly different running motor coordination compared to training matched runners, differences are observed between triathletes and novice runners (Chapman et al., 2008). Comparisons between cohorts from various sporting populations and triathletes could therefore explain the between study differences in kleg observed.

Average kankle (Equation J) for the current data was similar to results for sprint running (Kuitunen, Komi & Kyrolainen, 2002), but about half the stiffness of over ground running at a similar horizontal velocity (Joseph et al., 2013). The majority of literature reports knee stiffness to be higher than ankle stiffness in over ground running (Joseph et al., 2013; Kuitunen, Komi & Kyrolainen, 2002). The current cohort of triathletes showed equal group means for ankle and knee stiffness with many individual triathletes having lower knee than ankle stiffness. A similar relationship between kknee and kankle (Equation J) was reported for thirteen runners, running over ground at a range of velocities (Arampatzis, Bruggemann & Metzler, 1999). The footstrike pattern was not controlled in the current study’s cohort of triathletes. A higher ratio of midfoot and forefoot strikers in the subject population is likely to increase the average knee stiffness and decrease average ankle stiffness (Laughton, Davis & Hamill, 2003). Sprint running uses a predominantly forefoot strike, which could explain the similar joint stiffness between triathletes and sprinters despite different horizontal velocities (Kuitunen, Komi & Kyrolainen, 2002). However, further investigation into the relationship between knee and ankle stiffness and how inter-individual variation in factors like footstrike patterns influences this relationship is required.

The results, of the present study confirmed that vertical and leg stiffness have good inter-day reliability during running. Individual joint stiffness ranged from poor to good. While the measurement reliability (MDiff% and ES) were acceptable for ankle, knee and hip, the knee and hip had large measurement variability (%CV and ICC). When the joints were combined as sumjoints (Eq. (1)), hip +knee (Eq. (2)) or knee+ankle (Eq. (3)), variability was reduced to an acceptable level and overall reliability was improved to moderate to good. This result highlights the important role that regulation of joint stiffness holds in the co-ordination of lower body segemnts to achieve a task goal (Davids, Button & Bennett, 2008; Hamill, Palmer & Van Emmerik, 2012).

Stiffness control is initiated prior to ground contact with muscle pre-activation. The hip primarily controls the angle of the leg at contact and therefore the angle of the sweep of the leg. Less hip flexion would result in the centre of pressure at initial contact being closer to the centre of mass (Fig. 1), a smaller angle of the sweep of the leg and as a result less ‘leg spring’ compression, if all other factors remain equal. The knee allows adaptation to the environment following contact and therefore has greater variation than the hip and ankle (Lafortune, Hennig & Lake, 1996). The current results support control of the kinetic chain in this manner. Correlations of the hip and knee combination (r = 0.66) (Eq. (2)) and the three joints combined (r = 0.61) (Eq. (1)) with kleg∕time (Equation L2) were lower than for the knee and ankle combination (r = 0.82; 95% CI [0.52–0.94]) ((3)). Knee correlation with kleg∕time (r = 0.79; 95% CI [0.46–0.93]) (Equation L2) was similar to the knee and ankle combination (Eq. (3)) indicating that the knee is the primary controller of leg stiffness of triathletes running at the velocities assessed in the present study.

The current data suggests that in triathletes, hopping at 2.2 Hz, either with normal knee bend or a straight knee, does not correlate well with treadmill running for leg and joint stiffness estimates. Interestingly, bent knee hopping correlates moderately with running for leg stiffness but straight knee hopping has better correlations than bent knee for joint stiffness. Gait has been divided into two models based on the trajectory of the centre of mass, the ‘inverted pendulum’ for walking (Mochon & McMahon, 1980) and ‘bouncing/spring-mass’ for running (Blickhan, 1989; Cavagna & Kaneko, 1977; McMahon & Cheng, 1990). Like running, the centre of mass trajectory during hopping is lowest at the time point of highest vertical force and highest at mid-flight. However, projecting the centre of mass forward during running necessitates different muscle activity with changes in the electromyography profiles compared to vertical jumps or hops (Mero & Komi, 1994). During straight knee hopping, the ankle is the dominant joint controlling stiffness resulting in similar stiffness between the knee and ankle. Extending the leg anterior to the centre of mass during running results in a relatively small knee flexion compared to bent knee hopping (Farley & Morgenroth, 1999; Ferber, McClay Davis & Williams Iii, 2003) which could explain the better match for knee stiffness between straight knee hopping and running. Bent knee hopping however, is a more natural example of how all the tissues of the lower limb work together to absorb the impact of landing. Leg stiffness is a measure of this interaction within the lower limb system, which could explain the improved correlation between the running and bent knee hopping for leg stiffness.

Besides biomechanical differences in the task demands between hopping and running, the poor stiffness correlations between these two tasks could be due to the lack of familiarisation the triathletes had for hopping. It should also be noted that stiffness is altered as a result of changing contact time (Girard, Micallef & Millet, 2011; Girard et al., 2013; Hobara, Kanosue & Suzuki, 2007; Hoffrén et al., 2011; Morin et al., 2007). Therefore, the hop or stride frequency would also influence stiffness and the potential relationships between stiffness as measured during hopping and running. In a one hour fatiguing run, runners’ preferred stride frequency ranged from 1.36 to 1.60 Hz (Hunter & Smith, 2007). It is possible that hopping would provide more comparable stiffness estimates if the triathletes in the current study hopped at their preferred stride frequency. The study population contained a mixture of rearfoot and midfoot strike running patterns, based on presence or absence of a contact peak in the vertical ground reaction force, which may have also confounded the comparison between hopping and running. During hopping, landing generally occurs on the forefoot and therefore would be more likely to mimic a forefoot strike during running. The ankle angle profile of a rearfoot strike runner shows an initial plantarflexion action followed by dorsiflexion, which is not apparent in forefoot runners (Williams III, McClay & Manal, 2000). Future research should therefore look to separate the rearfoot and forefoot runners and/or assess stiffness as fractions of the first half of stance to determine whether this may provide better insight into the correlations between hopping and running in different sub-populations of running athletes.

Conclusion

Due to the superior reliability and correlations with both combinations of joint stiffness and vertical stiffness, the kleg∕time (Equation L2) method is recommended for assessing leg stiffness. However, kleg∕dynamic (Equation V2) also appears to be a good measure and when combined with joint angle recording also allows for calculation of joint stiffness. Joints should be assessed as a system in relation to each other rather than in isolation to gain acceptable reliability of the measure. The knee and ankle combination appears to be the most important when assessing changes in leg stiffness for triathletes running at the paces assessed in the present study. Hopping at 2.2 Hz is not a good substitute for running when estimating stiffness in triathletes. Further analysis is required to determine if hopping correlates with running when frequency is constrained to the triathletes, preferred stride rate or assessed in sub-groups of athletes who differ in their foot strike patterns or familiarity with vertical hopping.

Supplemental Information

Supplemental Information 1 Raw data—body mass normalised and unitless modified running and hopping stiffness

Click here for additional data file.

Additional Information and Declarations

Competing Interests

Author Contributions

Ethics

Data Availability

Justin W.L. Keogh is an Academic Editor for PeerJ. The authors declare that there are no other competing interests.

Anna V. Lorimer conceived and designed the experiments, performed the experiments, analyzed the data, prepared figures and/or tables, authored or reviewed drafts of the paper, approved the final draft.

Justin W.L. Keogh conceived and designed the experiments, analyzed the data, authored or reviewed drafts of the paper, approved the final draft.

Patria A. Hume conceived and designed the experiments, analyzed the data, authored or reviewed drafts of the paper.

The following information was supplied relating to ethical approvals (i.e., approving body and any reference numbers):

Ethical approval was obtained for all testing procedures from the Auckland University of Technology Ethics Committee (AUTEC) (AUTEC reference #11/94).

The following information was supplied regarding data availability:

The raw data are provided in a Supplemental Information 1.

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
