# Peer review of "Using stiffness to assess injury risk: comparison of methods for quantifying stiffness and their reliability in triathletes"

_PeerJ, doi:10.7717/peerj.5845_

## Round 0.1 · original submission · Major Revisions

Dear authors,

After assimilating the reviewers comments I believe that the manuscript will be acceptable for publication once your team address fully the comments provide by the reviewers.

·

Basic reporting

No comment.

Experimental design

No comment.

Validity of the findings

No comment

Additional comments

Only some minor points:

Line 138-139: Please provide a references for this sentence.
Line 248-250: Please provide a references for this sentence.
Line 250-251: Please provide a references for this sentence.
Line 345: Please reconsider the wording "...reliability to moderate to good.
Line 382: Please reconsider the wording "...other athlete adjust...".

Reviewer 2 ·

Basic reporting

The manuscript was very well written with clear logic. Authors used unambiguous, and professional English throughout. It is easy to read and understand the manuscript.

Introduction
The introduction is sufficient to illustrate the background in the work and relevant prior literature is appropriately referenced. It clearly explained the measurement of the vertical stiffness, leg stiffness and joint stiffness, and clarified the applicability of the three measurements of stiffness in research. The writing is going on smoothly and without obstacles for understanding.
Line 84-87: Would you please explain briefly what are the five risk factors associated with risk of Achilles injuries in running athletes and linked to changes in lower limb stiffness?

Methods
Line 188: What did TrainingPeaksTM monitor in athletes? How do you decide the effects of variations was minimized by utilizing TrainingPeaksTM?

Results
The results were presented well even there were plenty of data.

Discussion
The discussion compared the average stiffness value calculated with different equations with those in previous studies, and explained clearly why the values were different among studies. It also highlighted the good inter-day reliability of vertical and leg stiffness during running, and hopping and running stiffness did not correlate with each other well. The results have been fully discussed and well explained.

One problem is that in the introduction part, the first paragraph mentioned the overuse injuries especially the Achilles tendon injuries in triathletes, but no related contents to echo the content of this part later in the manuscript. It may be good to add some discussions about the utilization of the stiffness measurement in triathletes with overuse injuries here.

Table 2, 3, 4, 5 should be modified to three-line tables.

Experimental design

The research question was well defined, relevant and meaningful. There was a very good review about the effects of the three stiffness measurements on performance and injury and implications for training (Brazier et al., 2014, Strength and Conditioning Journal, 36(5),103-112), but research about the comparison of different stiffness measurements as well as their reliability when used in determination of injury risk was few. The rationale of the manuscript is very strong and the research finding should be very meaningful in the future investigation.

The study design is sophisticated and comprehensive. The authors selected top-level triathletes and conducted the study based on the ethical approval from the university ethics committee. The rationale of non-randomisation was clearly presented. Setting of the testing procedure as well as data analysis method were reasonable and were stated precisely.

The details of the methods were described very clear and organized.

Validity of the findings

Data is robust, statistically sound, and controlled.

Conclusion are well stated, linked to original research question and limited to supporting results.

Additional comments

In general, the study was well designed and conducted very carefully. The different methods of stiffness measurement were usually confusing in different studies. Therefore, it is imperative to clarify each of the methods as well as compare the utilization of them so that researchers could choose the proper method to investigate related research questions. This study has done a good job in distinguishing the methods of the stiffness measurement, reporting the reliability and correlations of the methods, and making the recommendation about the utilization of the methods under different situations. This will greatly contribute to the future research in assessing injury risk with stiffness measurement.

Reviewer 3 ·

Basic reporting

While the literature review has been well written, there has been several prominent pieces of work within stiffness literature which investigate this topic and have been overlooked in the literature review such as Diggin et al. 2016; Joseph e al; Pruyn et al. One article in particular has been cited for it's stiffness measures but the author does not discuss their reliability results. Tables are well presented. Some concepts also require a citation to support this e.g. line 111-112 which has been established in research. Once again 154-155 this concept has been explored particularly in hopping and comparisons between hopping stiffness and running stiffness have been made/investigated.

Experimental design

Overall there a several issues with the experimental design of this project, while there is a good research questions and has the ability to add to the body of research. There are a few flaws with the methods used, particularly the calculations which appear to be done without any justification or scientific back up e.g. the summing of joints. Summing of joint stiffness values appears to have been done to improve reliability, rather than with evidence to support the sum of the values e.g. line 171-173. The joints involved have varying degrees of freedom and the point of peak moment will vary as the joints time different e.g. as suggested in the paper the hip pre-activates first. Additionally transforming stiffness numbers to unitless values appears unnecessary and with little scientific support, suggest revisiting this concept. Would also suggest redefine this aim or presenting data on speeds that are reflective of elite triathletes as suggested by the study (which appears to be captured). Elite triathletes threshold pace is 3 min/km, 5 min/km pace would significantly influence your stiffness values and its comparison to hopping. This is noted in the conclusion of the article but is a contradiction to the what the paper suggests initially line 499 - 502. I suspect the population is a recreational group which would have an effect on the joint stiffness reliability as they potentially will utilize individual strategies and joint dominance or contributions may change. Would also suggest technique may vary significantly between Olympic level distance and long distance triathletes.

Validity of the findings

Further statistical analysis needs to be undertaken, it is stated that data was assumed to be normal, were normality tests undertaken? If not this will affect the statistical results. Additionally, PCA analysis could of been undertaken to assess which joints where the main contributors to movement. Methodology of the paper needs to be revisited or more clarity provided. This will effect the findings of results as it will provide between comparisons e.g. comparing stiffness levels in the discussion to previous papers, it is difficult to compare different population groups due to different training levels. Recreational athletes are very different to elite athletes, based on the running speeds, age and weight presented in this study it suggest they a recreational which will change how the data is discussed and what speeds should the data would be presented. Would rather see data presented on speeds reflective of their true competition pace.

Additional comments

The paper provides some good insights and has potential. It is well written and covers the main articles in this topic. Major revisions are required on the methodology of this paper, justification of methods around summation of joint stiffness values and the unitless values is required. This will change the interpretation of findings and conclusions drawn. Consideration needs to be given to the population recruited and taken into consideration into the interpretation of findings and further investigation into existing literature is required and will provide guidance and assistance is developing this paper.

---

## Round 0.2 · accepted · Accept

Thank you for your careful revision. We are glad to accept this submission.

Reviewer 2 ·

Basic reporting

no comment

Experimental design

no commen

Validity of the findings

no comment

Additional comments

After the revision, the rationale of this study has been more clearly and fully explained. Relevant prior literature has been appropriately referenced. Article structure is more complete.

Only one suggestion:
Line 551-552, line 632: please keep the format of the two references consistent with the rest of them.